# Does Spontaneous Secondary Succession Contribute to the Drying of the Topsoil?

**Edyta Hewelke** [1,*], **Piotr T. Zaniewski** [2], **Ewa Zaniewska** [3], **Ewa Papierowska** [1], **Dariusz Gozdowski** [4], **Andrzej Łachacz** [5] **and Ewa Beata Górska** [6]

1   Water Centre, Institute of Environmental Engineering, Warsaw University of Life Sciences, Nowoursynowska 159, 02-776 Warsaw, Poland

2   Department of Forest Botany, Institute of Forest Sciences, Warsaw University of Life Sciences, Nowoursynowska 159, 02-776 Warsaw, Poland

3   Department of Environmental Protection and Dendrology, Institute of Horticultural Sciences, Warsaw University of Life Sciences, Nowoursynowska 159, 02-776 Warsaw, Poland

4   Department of Biometry, Institute of Agriculture, Warsaw University of Life Sciences, Nowoursynowska 166, 02-776 Warsaw, Poland

5   Department of Soil Science and Microbiology, Faculty of Agriculture and Forestry, University of Warmia and Mazury in Olsztyn, Pl. Łódzki 3, 10-727 Olsztyn, Poland

6   Department of Biochemistry and Microbiology, Institute of Biology, Warsaw University of Life Sciences, Nowoursynowska 166, 02-787 Warsaw, Poland

\*   Correspondence: edyta_hewelke@sggw.edu.pl; Tel.: +48-225935356

**Abstract:** The aim of the study was to analyse the moisture content ($\theta$) and the persistence and strength of water repellency (SWR) on sandy soil excluded from cultivation and then undergoing spontaneous afforestation or weed infestation during an abnormally warm period. Three site plots in close proximity were selected, i.e., Scots pine forest—S1 (*Dicrano-Pinion*), birch forest—S3 (*Molinio-Frangulion*) 25 years old, and an abandoned field—S2 (*Scleranthion annui*) 1 year old, in Central Poland. The study covers the growing period in 2020 for the two upper soil layers. For the top layer, the average $\theta$ values for S1, S2, and S3 were 0.069, 0.101, and 0.123 cm$^3$cm$^{-3}$, respectively. In S1, the $\theta$ values were close to the permanent wilting point, and the actual SWR classes (water drop penetration time WDPT test) indicated the top layer as belonging to the extremely repellent class continuously for almost the whole study period. For other sites and soil layers, the wettable SWR classes were assessed. Whereas the severity of the potential SWR, based on measured values of the wetting contact angle (sessile drop method), also showed hydrophobicity for the top layer under the birch forest. The study provides new information regarding the risk of drying out the soil due to the SWR in sandy soils, depending on land use and climate warming.

**Keywords:** afforestation; soil hydrophobicity; bio-hydro-physical soil properties; sandy soil; soil moisture content

## 1. Introduction

The role of soil management in addressing climate change mitigation and adaptation is a crucial issue in many European expertise forums. Land abandonment, the most frequent reason for landscape change [1], is influenced by a complex range of drivers that vary over space and time [2]. The abandonment has occurred mostly in Eastern and Central European countries in the last few decades [3]. According to Ustaoglu and Collier [2], areas where the abandonment is supposed to be located are in Southern, Eastern, and Central Europe, parts of the UK and Ireland, and the Scandinavian countries.

Agriculture on low-productivity, sandy soils was largely abandoned as economically unjustified. The share of post-agricultural lands in State Forests in Poland is at least 22.1% [4]. These sites are predominantly covered by secondary pine forests. On the other hand, birch is among the most common species on recently abandoned farmlands [3].

There is an increasing need for the restoration of deciduous forests within coniferous forest plantations [5], as mixed forests in Poland are more resistant to climatic and environmental changes. Such swaps are possible even on oligotrophic, sandy sites [6]. The appropriate choice of species is important since the plant cover can change the water regime and influence the direction of hydrological processes connected with the topsoil [7]. Excluding land from agricultural use in the early years increases the soil's organic matter in both its active and passive fractions [8], which may increase soil water repellency (SWR) [9].

In forest habitats, the presence of SWR is recognised in both coniferous and deciduous forests [10–15]. Additionally, the SWR also appears on forest roads in semixeric and mixed coniferous forests [16]. In forest soil, the hydrophobic material is unevenly distributed [13] but accumulates mainly in the topsoil [11]. In pine forests, the SWR is connected with the presence of needles and bark rich in resins, waxes, and aromatic oils [17,18]. The strength of SWR depends on the pine species and may range from slight to strong [19]. In addition, Kajiura [20] proposed a stochastic approach based on the distributions of soil water content and SWR for the estimation of the areal fraction of the soil surface exhibiting water repellency for various soils in a humid-temperate forest, which explained 70% of the variability.

The value of soil moisture is a relevant factor that can induce or prevent the formation of a repellent soil layer. The soil is least repellent or non-repellent when moist and most repellent when dry [21,22]. The predicted increase in annual mean global temperature and heatwave frequency is expected to extend the spatial extent and severity of SWR and the related soil-water balance processes [23]. The repellency phenomenon changes the soil's functional behaviour across a soil-specific range of water content. In turn, they contribute to reducing the amount of water available to plants, and may significantly increase surface runoff [24] and nutrient loss [25], as well as facilitating soil erosion [26,27]. Drying out of the topsoil also poses a fire hazard [28]. In addition to many negative effects of SWR, its positive effects are often emphasised [29], e.g., its influence on carbon sequestration mechanisms [30]. On the other hand, Urbanek and Doerr [31] and Sánchez-García et al. [32,33] showed that SWR can become a key indicator influencing $CO_2$ release, especially during rainfall events.

Schlesinger et al. [34] recommend that drought in nature requires better recognition, and in turn, the ability to manage forests in the face of drought and the parameterisation of drought in earth system models can improve predictions of carbon uptake and storage in the world's forests. Huang and Hartemink [35] drew special attention to sandy soils, which are more responsive to climate change and human pressure on land resources, and also recommended improving models to describe water and nutrient transport.

The aim of the study was to analyse the soil moisture content ($\theta$) and the persistence and strength of SWR on sandy soil (Albic Podzol) excluded from cultivation and then undergoing spontaneous afforestation or weed infestation. The abnormally warm growing period was chosen, and the study was conducted on three different cover sites in Central Poland. We hypothesised that during the warm period, plant cover can significantly change the soil moisture content in the upper soil layers. The cause of that change is the occurrence of soil hydrophobicity phenomena, and two independent methods were used to assess the SWR.

## 2. Materials and Methods

### 2.1. Study Area

The study area is located near Stanisławów village in the Mazovian lowlands (Central Poland). The geological substrate of the site is aeolian sand [36]. According to the Köppen—Geiger classification [37], the climate is warm temperate (Cfb), with an average annual temperature of 7.8 °C and an annual rainfall of 545 mm. The study area (ca. 13.46 ha) is flat, and the only significant denivelation is a small dune ridge (circa 2 m in height) located in the north part. Due to the low productivity of the soil, most of the area, which had

been used to grow mainly potatoes and rye, was abandoned for agricultural production in the 1990s and left to spontaneous secondary succession (Figure 1a–c).

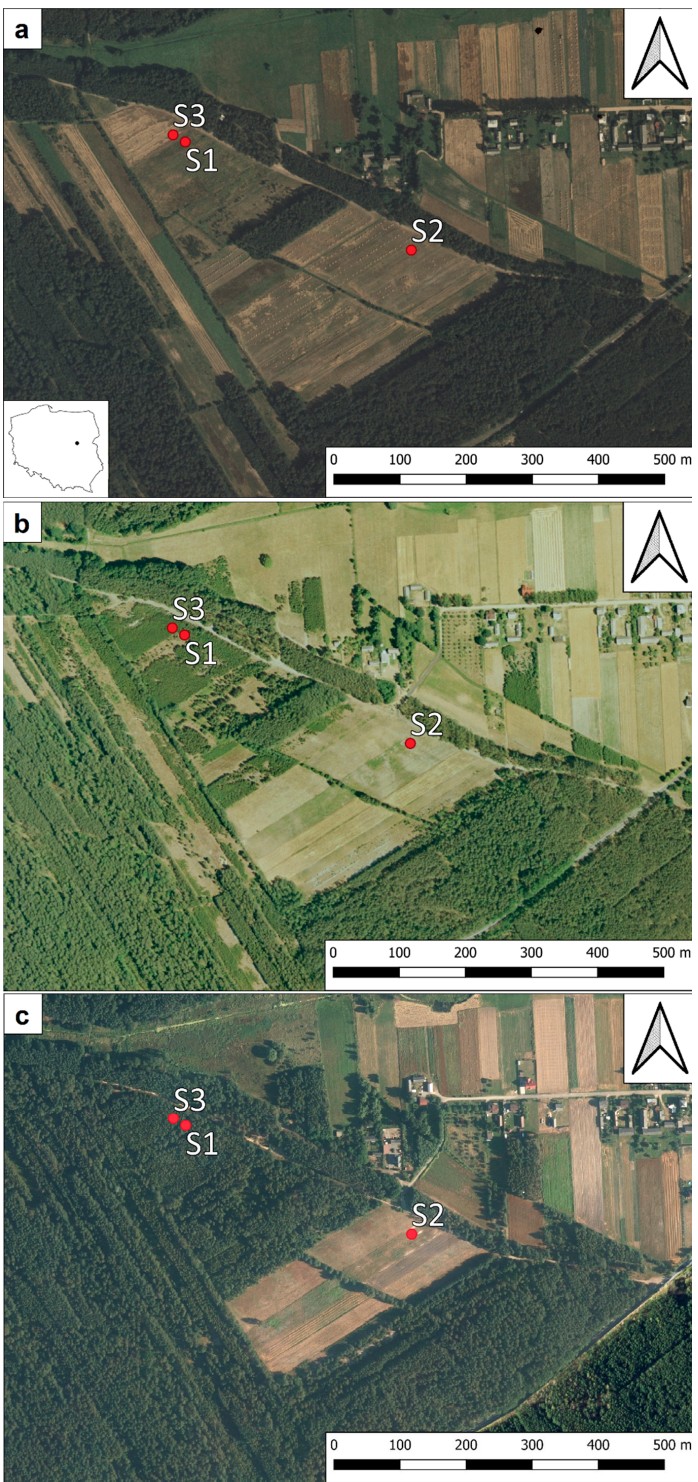

**Figure 1.** Location of the study sites: *Dicrano-Pinion* S1, *Scleranthion annui* S2, and *Molinio-Frangulion* S3, with the historical land uses, (**a**)—1997, (**b**)—2006, and (**c**)—2018. Source of orthophotomaps: Geoportal [38].

The three orthophotomaps, (a) from 1997, (b) from 2006, and (c) from 2018, present the historical land use change in the last two decades (Figure 1). In 1997, according to our own elaboration, ca. 84% of the area was agriculturally used, 8% was already abandoned,



and 8% was overgrown by young forests. In 2006, only 42% of the land was managed, with 20% abandoned and 38% overgrown. In 2018, 35% of the site was still used, 5% was not used, and 60% was already covered by trees. Another field (4% of the area) was abandoned during the year of the study. Silver birch (*Betula pendula* Roth) and Scots pine (*Pinus sylvestris* L.) are the dominant tree species in the recently formed initial forest vegetation. Three representative site plots in close proximity (maximum distance 400 m) were selected (Figure 1), i.e., a Scots pine forest, 25 years old (S1, WGS84: 52°17′07.9″ N, 21°31′01.0″ E), an abandoned, 1 year old (S2, WGS84: field 52°17′02.9″ N, 21°31′19.6″ E), and a birch forest, 25 years old (S3, WGS84: 52°17′08.8″ N, 21°31′01.9″ E).

The study covers the growing period from April 2020 to September 2020. The previous winter was snowless, and at the beginning of April, the water level of the main rivers was between the medium and low water zones. The spring period was classified as dry, the summer as wet, and the analysed period as abnormally warm compared to the multi-year period 1981–2010, according to IMGW-PIB [39]. Even winter's monthly average air temperature values were above zero, and May was the coldest month of 2020. The year's classification as abnormally warm was most influenced by June, when the average temperature was higher by 2.2 °C compared to the multi-year data in this region. There was a considerable variation in terms of precipitation in individual months. According to the multi-year classification, March— the month preceding the research period and July were very dry months, and the extremely dry month was April. The wet months were May and September; June turned out to be extremely wet, while August was considered a normal month. The daily detailed data of rainfall and temperatures from the nearest station in Siedlce (52°03′40″ N 22°33′2″ E) for the analysed growing period, are presented in Figure 2a. The impact of land use changes and plant cover was analysed in two topsoil mineral layers (after removing litter). The main properties, defined in the middle of the upper two mineral soil horizons of the Albic Podzols based on the World Reference Base—WRB [40], are presented in Table 1. The particle size distribution was analysed by the sieve and hydrometric methods [40,41]. Soil moisture retention characteristics were measured in a laboratory in triplicate on undisturbed soil samples (100 cm$^3$) using a reference method [42]. The moisture content values of pF between 0.4 and 2.0 were determined in a standard sand box, whereas the amounts of water at pF 2.3, 2.7, 3.4, and 4.2 were measured in pressure chambers, and details are presented in a previous study by Hewelke [43]. Soil pH was measured potentiometrically in 1M KCl suspensions in the ratio of (equivalent soil: solution) 1:2.5. Total organic carbon (TOC) and total nitrogen (TN) contents were measured with a Vario Max Cube CN Elementar analyser. The TOC content, as well as the C/N ratio and pH, are balanced within the studied sites, which indicates post-agricultural soils that were cultivated and fertilised for a long time in the past.

**Table 1.** Main properties of the top soils at the three study sites.

| Characteristic | S1 | | S2 | | S3 | |
|---|---|---|---|---|---|---|
| Depth [cm] | 0–5 | 35–40 | 0–5 | 35–40 | 0–5 | 35–40 |
| Textural classes | Fine sand (97; 2; 1) [1] | Fine sand (97; 2; 1) [1] | Fine sand (95, 3; 2) [1] | Fine sand (97;3; 0) [1] | Fine sand (97; 2; 1) [1] | Fine sand (97; 2; 1) [1] |
| Bulk density (kg/m$^3$) | 1310 | 1476 | 1317 | 1417 | 1319 | 1552 |
| Full water capacity (cm$^3$cm$^{-3}$) | 0.422 | 0.423 | 0.437 | 0.435 | 0.505 | 0.436 |
| $\theta$_pF 2.0 (cm$^3$cm$^{-3}$) | 0.204 | 0.190 | 0.205 | 0.192 | 0.198 | 0.191 |
| $\theta$_pF 4.2 (cm$^3$cm$^{-3}$) | 0.070 | 0.072 | 0.070 | 0.073 | 0.105 | 0.072 |
| pH (KCl) | 3.93 | 4.65 | 3.78 | 4.61 | 3.78 | 4.61 |
| TOC (%) | 1.995 | 0.145 | 2.186 | 0.186 | 2.071 | 0.155 |
| TN | 0.180 | 0.012 | 0.150 | 0.015 | 0.144 | 0.012 |
| C/N ratio | 11.08 | 12.08 | 14.57 | 12.40 | 14.38 | 12.91 |

[1] The percentage of each main fraction [%]: sand, 2–0.063 mm; silt, 0.063–0.002 mm; clay < 0.002 mm [40].

### 2.2. Phytosociological Sampling

Vegetation sampling was conducted in July 2020, using the relevé method of Braun-Blanquet's [44] approach and the Barkman et al. [45] scale. Cover was used as a measure of species abundance, as recommended in vegetation surveys [45,46]. The area of each relevé was 100 m$^2$. Vegetation units were described according to the recent European reference classification of vegetation [47]. Additionally, the changes in vegetation were assessed with orthophotomaps of the site from 1997, 2006, and 2018 (Geoportal [38] presented in Figure 1a–c).

### 2.3. Soil Sampling and Soil Moisture Measurements

During the 2020 growing period (April–September), biweekly systematic measurements of $\theta$ were carried out for the two upper layers of soil at the depths of 0–5 cm and 35–40 cm. At each site and sampling date, three undisturbed samples of 100 cm$^3$ were collected, in close (up to 1 m) parallel transects. At Sites S1 and S3, the sampling points were roughly representative of the distance between trees. In 12 sampling campaigns on the 3 sites in the middle of 2 upper soil horizons in 3 replicates, 216 soil samples were collected to determine the volumetric $\theta$ using the gravimetric method by drying in the oven at 105 °C. The mean values from 3 determinations in each campaign and soil layer were used for statistical analysis.

### 2.4. Soil Water Repellency Assessment

Actual SWR was classified as the median value of the water drop penetration time (WDPT) test, for which a detailed description of measurements was presented by Hewelke et al. [48]. The test was conducted on 72 soil samples (12 campaigns × 3 sites × 2 soil layers) with actual field moisture. According to the SWR classification [49], 5 s was the limit between wettable and non-wettable (repellent) soil.

The severity of potential SWR was determined for the top 0–5 cm layer based on the wetting contact angle (CA) values. Contact angle measurements were conducted using a CAM 100 optical goniometer (KSV Instruments, Finland) at constant laboratory temperature (20 °C) and relative humidity ranging from 30% to 45%. The soil samples were air-dried and homogenised. The detailed measurement procedure is described in Papierowska et al. [50]. Measurements were performed consecutively at 1 s intervals for 120 s and repeated 13 times (13 drops). The values of CA were measured from the right and left drop sides for each drop ($n = 26$), and for further analysis, the values from the first 5 s were calculated. A wetting CA value equal to 90° was taken as the limit between wettability and non-wettability [51].

### 2.5. Statistical Analysis

The means of soil moisture content following normal distribution were compared among the experimental sites by analysis of variance; homogeneous groups were distinguished by the Tukey test for $\alpha = 0.05$ using the Statgraphics Centurion version XVI programme (StatPointTechnologies, Inc., 2009, Warrenton, VA, USA). Medians with quartiles for variables that were not followed a normal distribution (WDPT test) or means with standard deviations for variables that followed a normal distribution (wetting CA) were calculated. For non-parametric statistical analysis, the Kruskal–Wallis test was used for comparisons between the experimental objects for SWR analysis.

## 3. Results and Discussion

### 3.1. Spontaneous Vegetation Succession

The vegetation types of the surveyed sites represented three syntaxonomic units (Table 2). The first site (S1) represented a young, spontaneous pine forest from the *Dicrano-Pinion sylvestris* (Libbert 1933) and W. Matuszkiewicz 1962 alliance. Scots pine dominated the stand; however, a small admixture of birch was visible. The herb layer was poorly developed, but graminoids (*Agrostis gigantea* Roth and *Holcus mollis* L.) were characterised

by slightly higher abundance. The specific feature of the site was the developing bryophyte layer, with typical species of coniferous forests, such as *Hylocomium splendens* (Hedw.) Schimp. and *Pleurozium schreberi* (Willd. ex Brid.) Mitt. The second site (S2) was the recently abandoned field from *Scleranthion annui* (Kruseman et Vlieger 1939) Sissingh in Westhoff et al., 1946 alliance. It was dominated by grasses, including the minor weed of arable fields, *Antoxanthum aristatum* Boiss. The tree, shrub, and bryophyte layers were not developed. The third site (S3) was the young birch stand belonging to *Molinio-Frangulion* Passarge in the Passarge et G. Hofmann 1968 alliance. Its stand was dominated by birch, the shrub layer by alder buckthorn (*Frangula alnus* Mill.), and the herb layer by *Juncus effusus* L. and *Rubus* sect. *Rubus*. The bryophyte layer was poorly developed.

**Table 2.** Vegetation characteristics of the studied sites.

| No. of Study Sites | S1 | S2 | S3 |
|---|---|---|---|
| Area of relevé [m$^2$] | 100 | 100 | 100 |
| Cover of tree layer [%] | 65 | 0 | 50 |
| Cover of shrub layer [%] | 3 | 0 | 20 |
| Cover of herb layer [%] | 8 | 90 | 40 |
| Cover of bryophyte layer [%] | 20 | 0 | 2 |
| Vegetation type | *Dicrano-Pinion* | *Scleranthion annui* | *Molinio-Frangulion* |
| Tree layer | | | |
| *Betula pendula* Roth | 2a | | 2b |
| *Betula pendula* × *B. pubescens* | | | 3b |
| *Pinus sylvestris* L. | 4a | | |
| Schrub layer | | | |
| *Betula pendula* × *B. pubescens* | | | 1 |
| *Frangula alnus* Mill. | 1 | | 2a |
| *Pinus sylvestris* L. | | | 1 |
| *Salix cinereal* L. | r | | + |
| *Sorbus aucuparia* emend. Hedl. | | | 1 |
| Herb layer | | | |
| *Agrostis capillaris* L. | | | + |
| *Agrostis gigantea* Roth | 1 | 3a | |
| *Antoxanthum aristatum* Boiss. | | 2a | |
| *Carex ovalis* Gooden. | r | | |
| *Elymus repens* (L.) Gould | | + | |
| *Frangula alnus* Mill. | + | | + |
| *Holcus mollis* L. | 1 | 3b | |
| *Juncus effusus* L. | + | | 2a |
| *Lysimachia vulgaris* L. | | r | r |
| *Quercus robur* L. | + | | + |
| *Rubus* sect. *Rubus* | | | 2b |
| *Rumex acetosa* L. | | r | |
| *Rumex acetosella* L. | r | | |
| *Solidago virgaurea* L.s.str. | r | | |
| *Sorbus aucuparia* | | | + |
| Bryophyte layer | | | |
| *Brachythecium rutabulum* (Hedw.) Schimp. | | | + |
| *Cladonia glauca* Flörke | r | | |
| *Cladonia fimbriata* (L.) Fr. | r | | |
| *Dicranum polysetum* Sw. ex anon. | + | | |
| *Dicranum scoparium* Hedw. | r | | |

**Table 2.** *Cont.*

| No. of Study Sites | S1 | S2 | S3 |
|---|---|---|---|
| *Hylocomium splendens* (Hedw.) Schimp. | 1 | | |
| *Sciuro-hypnum oedipodium* (Mitt.) Ignatov & Huttunen | r | | + |
| *Pleurozium schreberi* (Willd. ex Brid.) Mitt. | 1 | | r |
| *Pohlia nutans* (Hedw.) Lindb. | r | | |

Cover values (%) of alphanumeric symbols: 4a = 50%–62.5%, 3b = 37.5%–50%, 3a = 25%–37.5%, 2b = 12.5%–25%, 2a = 5%–12.5%, 1 = 1%–5%, + = 0.5%–1%, r = 0.01%–0.5%.

Stands located within the same initial soil conditions but differing in the time of abandonment can be treated as part of the chronosequence [52]. The succession on abandoned fields usually starts with segetal vegetation, which then changes into grassland and scrubs or directly to forest communities [53]. However, many alternative succession trajectories are possible even within a single site, including gradual vegetation changes or direct forest tree species encroachment [54]. The surveyed vegetation types represented two successional series of oligotrophic, acidophilous habitats, frequent in the lowlands of Poland [53]. The birch encroachment is commonly observed during spontaneous succession on abandoned agricultural sites [3,55,56]. The birch is a pioneer species due to its large production of light seeds that are wind-carried over long distances. Birch-dominated habitats from the *Molinio-Frangulion* alliance are a feature of the agricultural landscape and are one of the transitional stages of secondary succession towards mixed oak-pine or acidophilous oak forests [57]. However, encroachment of both Scots pine and silver birch is possible during secondary succession, and these species can swap depending on the properties of the local habitat or the past land-use type [58] or the distance from seed sources. As a result, many post-arable lands are covered by pine forests [59]. A similar situation was seen at the study site. The patches of abandoned fields are dominated by birch, pine, or a mixture of these two species. The reason for such differentiation could be the slightly different management of the studied arable fields at the time of their abandonment. This can be manifested by other tones visible within the abandoned fields on the orthophotomap from 1997 (Figure 1a), the present-day birch and pine-dominated sites.

*3.2. Soil Moisture Dynamics*

Against the same background of the daily distribution of the rain events, the distribution of the average values of $\theta$ measurements in the two upper layers among the three sites was lowest at S1—early-successional pine forest, *Dicrano-Pinion* (Figure 2). The values were close to the permanent wilting point during the entire growing period. Even a wet May and an extremely wet June did not change this tendency. Observations from the previous year, March 2019–March 2020 [48], also showed low values of $\theta$ for S1. The persistence of low $\theta$ over a long period of time in these layers leads to permanent changes related to the transformation of organic matter. The in-depth characterisation of SOM by fluorescence spectroscopy [60] showed that land use significantly influences the direction of the SOM humification process. Almost three decades of transformation do not cause changes in the elemental and/or fractional composition of humic substances but result in a lower number of humic-like structures in humic acid, causing changes in the utilitarian properties of the soil.

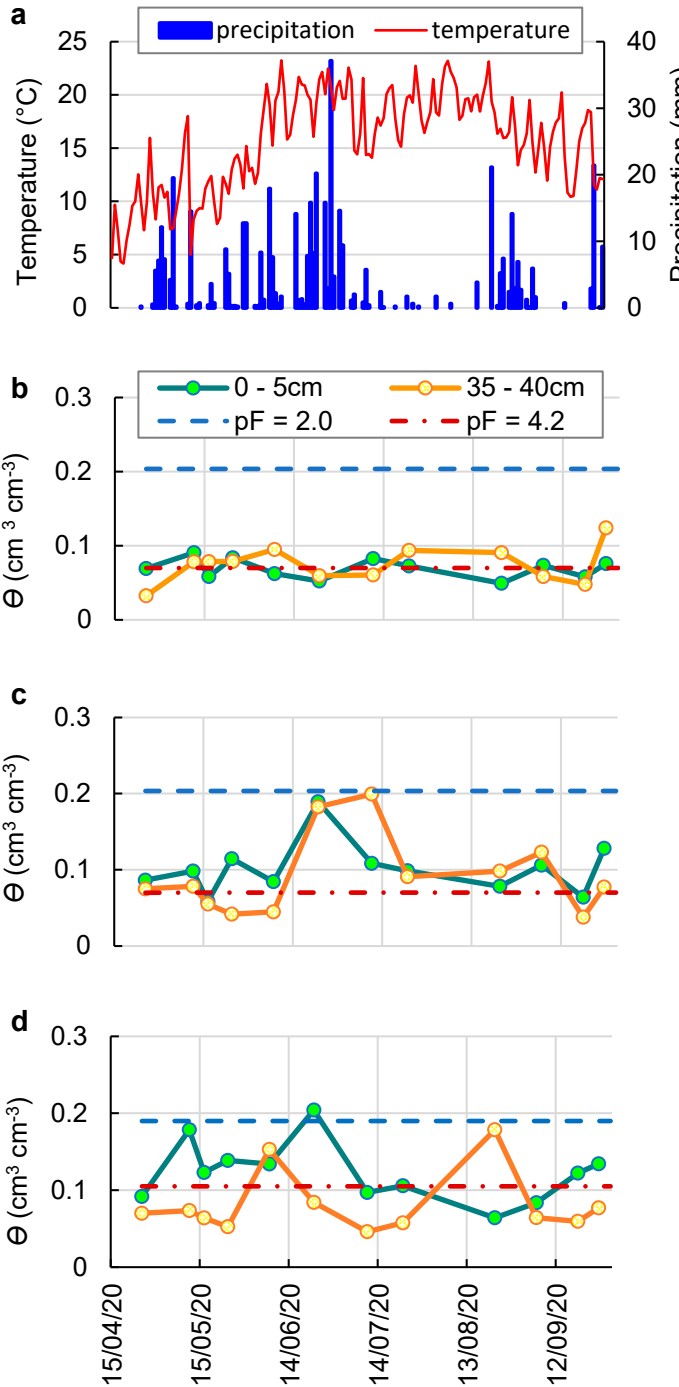

**Figure 2.** The sum of daily precipitation and mean daily temperature recorded at the nearest meteorological station (**a**) the mean soil moisture content (*θ*) for two mineral layers of soil under different plant cover, ((**b**) S1—*Dicrano-Pinion*, (**c**) S2—*Scleranthion annui*, and (**d**) S3—*Molinio-Frangulion*) from April, 2020 to September, 2020. Note: presented pF values are for 0–5 cm soil layer.

Whereas, in the cases of S2 and S3, the distributions of *θ* dynamics were typically a consequence of the daily distribution of the rain events, and as a result of cumulative rainfall, moisture values were around pF = 2.0 in June (Figure 2c,d). The values of moisture content in the topsoil layers of 0–5 cm (mean 0.069, 0.101, and 0.123 $cm^3 cm^{-3}$, respectively for S1–S3) confirm significant differences (Figure 3). It should be emphasised that the highest level of moisture content at the early successional birch forest, *Molinio-Frangulion*, S3, as well as the highest value of *θ* at pF = 4.2, turned out to be 50% higher in comparison

to the other two sites, which is evidence of the different hydrological regimes. It is also worth noting that the observed changes in the upper soil layer occurred at objects S1 and S3 in truly close proximity, the distance between the measurements did not exceed 25 m.

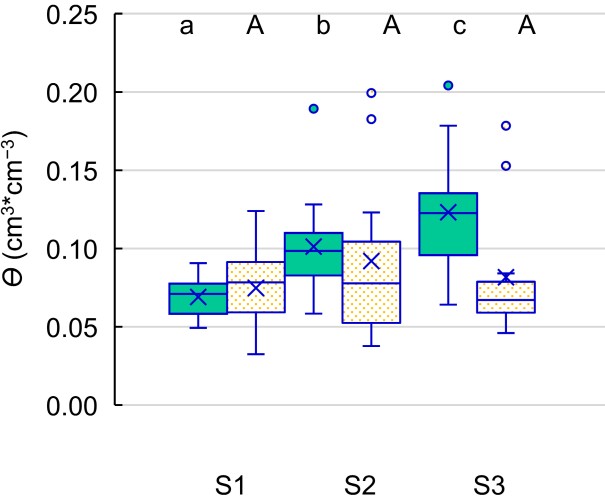

**Figure 3.** Box plots for soil moisture content (*θ*) for the 0–5 cm layer (green-filled boxes) and 35–40 cm layer (yellow boxes) of sandy soil in the three different plant covers: mean values (X), medians (horizontal lines inside boxes), the first and third quartiles (boxes), outliers (dots). Note: different letters denote significant differences in sites (the lowercase/uppercase letters distinguish between layers), as determined with the Tukey test (*α* = 0.05).

In the 35–40 cm soil layer, the *θ* values have no significant differences, and the average values for this layer were 0.074, 0.097, and 0.085 $cm^3 cm^{-3}$, respectively. However, the abandoned field, *Scleranthion annui* (S2) site was characterised by the largest scatter of values, which confirms that the water from the above layer was the easiest to move into the depth of the soil profile.

### 3.3. Soil Hydrophobicity

The soil repellency presented as the median of the WDPT results for all measuring campaigns shows significant differences (Figure 4). The hydrophobicity for *Dicrano-Pinion*—S1, 0–5 cm layer (Figure 4a) was identified as extremely repellent class (WDPT > 3600 s) based on the Dekker and Jungerius SWR classification [49], while the values of the lower quartile belong to the severely repellent class (600 s < WDPT ≤ 3600 s), whereas the values for S2 and S3 were characterised as wettable class (WDPT ≤ 5 s). The hydrophobicity phenomenon exists for some soils within a specific range of moisture content [61]. The threshold water content range is the transition zone, after exceeding which hydrophilic soils become hydrophobic [62]. In earlier studies under laboratory [43] and field conditions [48], a critical threshold of soil moisture was determined for S1 at a level of 14–16 vol. %. The low levels of moisture on S1 during the study period, was driven by non-wettability, the processes are in feedback relation, leading to the deepening of soil drought in the top and deeper soil layers (35–40 cm) (Figures 3 and 4a); however, all deeper layers were assessed as hydrophilic (Figure 4b).

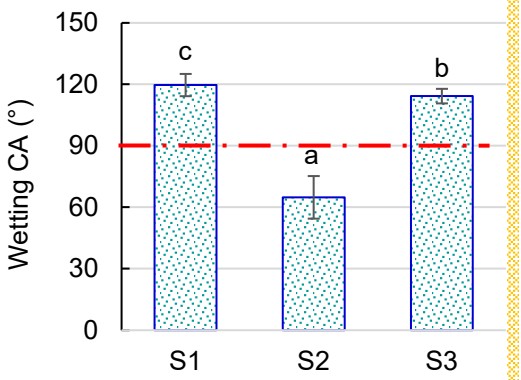

**Figure 4.** The median of the water drop penetration time (WDPT) test with upper and lower quartiles (range of quartiles is equal to 1 for all sites except S1, 0–5 cm), as actual soil water repellency (SWR), on the three sites, in two soil depths of (**a**) 0–5 cm and (**b**) 35–40 cm. Note: The red dashed line, WDPT = 5 s, is the boundary between wettable and water-repellent soil in the SWR classification (Dekker and Jungerius, 1990). Different letters denote significant differences between sites, as determined with the Kruskal–Wallis test ($\alpha = 0.05$).

The significant differences in the average wetting CA values obtained from the top mineral layer for the three study sites are shown in Figure 5. In the cases of S1 and S3, the average wetting CA was higher than 90°. This means that, in those cases, potential soil hydrophobicity was observed.

**Figure 5.** The mean of the wetting contact angle (wetting CA) with standard deviation as the potential SWR for the 0–5 cm soil layer for the three sites. The red dashed line, the wetting contact angle = 90°, is the boundary between wettable and water-repellent soil. Note: different letters denote significant differences between sites, as determined with the Kruskal—Wallis test ($\alpha = 0.05$).

For the pine forest, *Dicrano-Pinion* (S1), the wetting CA average of the topsoil (119.6°) was similar to that reported in the literature [11,63,64]. Both methods used in this study confirmed the presence of hydrophobicity for 0–5 cm in S1. According to Doerr et al. [17], in coniferous forests, soil water repellency can be a natural soil characteristic. Soil repellency under pine forests can be explained by the dropping of needles covered with hydrophobic waxes, resins, or aromatic oils in the autumn season, followed by the transformation of organic matter and rhizosphere activity. The high strength of SWR was observed at the pine-dominated sites in the Mediterranean region [65–67], as well as in Central Europe [10,19,68–71]. The persistence of SWR in sandy soils is changeable throughout the year and reaches its maximal strength in hot and dry summer or autumn seasons [10,48,72,73]. The long-lasting persistence and strength of SWR in the present research regarding warm temperate climate, which was even longer than in a wet Mediterranean

climate [67], may pose a meaningful environmental threat, especially to the regulatory ecosystem services.

In the case of S2, the abandoned field—*Scleranthion annui*, the soil was assessed as being hydrophilic (64.8°), which was confirmed by the two methods; it should be assumed that it will not disturb the hydrological function of the soil system.

For the birch forest, at *Molinio-Frangulion* (S3), a high mean value of the wetting CA equal to 114.2° was measured, although the soil hydrophobicity was not confirmed by the WDPT test. This may be due to the measurement conditions. The measurements of CA were carried out on air-dry soils, as opposed to the WDPT test, which was performed under field conditions in actual soil moisture. Generally, in the investigation period, the median WDPT values were equal to 1 s, and in this habitat, the highest $\theta$ values were recorded. The only exception is one campaign, on August 22, when the lowest $\theta = 0.0642$ cm$^3$cm$^{-3}$ was reached, with the median WDPT of 19,800 s—classified as an extremely repellent SWR class—which may indicate the transgression of the threshold water content zone for hydrophobicity. Lamparter et al. [74] noticed that, even in the air-dry samples, the results obtained by the two methods (WDPT and CA) were not compatible. The samples with a CA of about 90–100° were mostly associated with WDPT less than 2 s, which was explained by subcritical water repellency. The high values of CA measurements obtained for *Molinio-Frangulion*—S3 are surprising and require further investigation. Kajiura [20] also points out the role of the habitat condition with regard to the proposed protocol, which cannot be directly used to estimate the SWR areal fraction in places where soils are consistently wettable.

## 4. Conclusions

The plant cover has significantly influenced the soil moisture content in the top mineral layers during the warm vegetation period in 2020. In the case of S1 pine forest (*Dicrano-Pinion)*, the average $\theta$ value was close to the permanent wilting point in the two mineral top layers, while in the birch forest—S3 (*Molinio-Frangulion)* habitat—it was twice as high in the observation period.

The severity of the potential SWR, based on the measured values of the wetting CA, does not reflect the hydrophobicity occurring in the field conditions.

The WDPT test measurements of the actual SWR indicate that spontaneous afforestation with pine significantly strengthens the SWR phenomenon. Although it is laborious and time-consuming to estimate the critical threshold (repeated measurements of SWR and $\theta$), it may be that the soil property is fundamental for identifying the occurrence of risks connected with the drying out of the topsoil. Maintaining high levels of soil moisture, as well as mixed forest afforestation against the dominant pine afforestation, can turn climate and environmental challenges into opportunities to mitigate and adapt to the changes. Considering the water deficit, the strategy connected with mixed forest afforestation requires interdisciplinary commitment and special attention based on an understanding of the environmental conditions and processes that shaped the topsoil moisture regime. Multifunctional and healthy forests are increasingly socially desirable [5,75,76], and there is a need for further research on the interactions of climate, land use, and soil changes.

**Author Contributions:** Conceptualization, E.H., P.T.Z. and E.Z.; methodology, E.H., P.T.Z., E.Z. and E.P.; formal analysis, D.G.; investigation, E.H., P.T.Z., E.Z., E.P. and A.Ł.; writing—original draft preparation, E.H., P.T.Z. and E.P.; writing—review and editing, E.H. and E.P.; visualization, E.H. and P.T.Z.; supervision, A.Ł. and E.B.G. All authors have read and agreed to the published version of the manuscript.

**Funding:** This research received no external funding.

**Informed Consent Statement:** Informed consent was obtained from all subjects involved in the study.

**Data Availability Statement:** The detailed data presented in this study are available on request from the corresponding author.

**Conflicts of Interest:** The authors declare no conflict of interest.

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
