# Peer review of "Does Spontaneous Secondary Succession Contribute to the Drying of the Topsoil?"

_forests, doi:10.3390/f14020356_

Round 1

Reviewer 1 Report

The manuscript deals with the problem of moisture conditions of the topsoil layer (during growing season) in spontaneously forested post-agricultural land, with coniferous or deciduous tree stands and early succession vegetation. In different countries programs of afforestation of post-agricultural lands (also building field shelterbelts, developing agroforestry practices etc.) are developed and subsidised.  It can be observed (e.g. in Poland) that many of such lands are turned to monocultures of pine, mainly on poor, sandy soils. The paper is an interesting contribution to research on how to properly design tree stands in agricultural areas and to create some recommendations for the species selection, taking into account some risks (e.g. the risk of fires in dry periods under pine stands or biodiversity loss). An important element that would enrich the knowledge in this field would be the study of moisture and hydrophobicity of the organic layer/litter (especially during dry season) and including evapotranspiration measurements, which I hope the Authors will consider in further studies.

Some comments and recommendation for Author:

Soil properties were analysed in arbitrarily designated layers of specific depth (0-5 and 35-40 cm), not in “soil horizons” (which are reflecting soil processes, as in soil science). Therefore, in this sense, “horizon” should be changed to “layer” in the whole manuscript.

Authors have numbered the three survey sites: S1, S2, S3. I would advise to use this names consistently in the whole manuscript – text, tables, figures and figures captions  (instead of e.g. 1, site 1, Site 1).

91: what do Authors mean by “hydrological regime”? The soil hydrological regime as such was not studied here, but some of its elemnets in the topsoil layer.

97: the area of what exactly is given in brackets (13.46 ha)? Some justification for why this exact area of 13,46 ha was interesting for investigations and what sites S1, S2 and S3 represent should be explained here. Perhaps the sentence (128-131) “Three site plots in close proximity (maximum distance 400 m) …. 25 years old (S3, WGS84: 52º17’08.8”N, 21º31’01.9”E)” should be moved to line 103, before Figure 1.

107 - 113: The sentence “The three orthophotomaps …. present the historical land use change in the last decades” should be moved to the beginning of the paragraph. In this paragraph source of the numbers quoted need to be cited, so your own elaboration.

115-126: Is the pluvial and thermal classification of specific months accomplished based on of the data form the meteorological station in Siedlce?

133: I would emphasize that litter (or organic layer) was removed, e.g. ….mineral soil layer (after removing of litter)

135: To avoid confusions, the source of particle-size fractions and soil texture classification should be cited (FAO-UNESO…), as this classification is not widely used for agricultural soils (also sandy) in Poland.

137: The method used to determine soil water retention characteristics should be called (pressure plate method, centrifuge or other) and also what characteristic points were determined (pF value and name).  

139: …. 1:2,5 (m/v); “Corg.” change (everywhere) to “TOC”  and “Ntot” to “TN”. Was TOC determined directly in the analyser or calculated based on TC  (measured in the analyser) and inorganic C?

46: Information about dates of vegetation survey is lacking (month, year)

154: “Systematic measurements…” - what was the study period (it can be repeated here), intervals of the measurements and dates of the 12 campaigns?

159: … using gravimetric method, by drying….…

160 and 166: I’d rather move the information which parameters are normally distributed and which are not to section 2.5.

175: With the sampling time and sample numbers (n=26), it would be more clear to say – as previously -  that wetting contact angle was measured in one layer (0-5 cm) in each of 12 campaigns.

191: Taxonomy units and plant species names are presented in details, except for Agrostis gigantea and Holcus mollis, why?

202: Explanations of scale for coverage of species should be put under the Table 2. The most widely known is Braun-Blanquet scale, so it would be helpful for readers to have the scale of Barkman et al. explained.

221: “different management ….at the time of abandonment”… Do you mean “before they were abandoned”?

230: “low θ over a long period of time in these layers leads to permanent changes related to the transformation of organic matter”. This suggests that Authors were investigating SOM transformation at the study sites. Just remove “in these layers” from the sentence.

Figure 3: I think it should to be graphically improved

265: Scleranthion annui site (S2)

Figure 4: In the figure caption, full names of WDTP and SWR should be given with abbreviations in brackets; (and also instead of Site 1 – S1 etc.)

Figure 5: In Figures 3 and 4 abbreviations were used in the titles of axis, so this format should be used consistently. In the figure caption: “The mean of ….. with standard deviation…” (there is no need to use abbreviation here), full names with abbreviations in brackets e.g. wetting contact angel (wetting CA).

312: Please quote the exact value of wetting CA also for S2

319: “that period”, better “investigation period”

348: instead of “shaped hydrological regime” I would rather use “shaped soil/topsoil moisture conditions”

I don’t feel much of an expert in English, however I have put some comments on language, as well as some minor errors (e.g. editorial), in the manuscript.

Author Response

Dear Reviewer, 

All comments and suggestions were taken into account and applied to the text, and are visible in tracked version. The detailed answers are attached.

Best regards,

Reviewer 2 Report

Dear Authors

Manuscript title: Does Spontaneous Secondary Succession Contribute to the Drying of the Topsoil? 

It is an interesting work dealing with the contribution of Secondary Succession to topsoil drought by considering soil moisture and water repellency. Overall, the manuscript is logically organized and well illustrated, and the interpretations are well supported by the data. 

I added minor comments to the attached file to improve the quality of the manuscript.

Author Response

Dear Reviewer, 

All detailed comments and suggestions were taken into account and applied to the text, and are visible in tracked version. The values of soil moisture content in the previous version of the text were presented as %, thanks to accurate remarks the oversight was corrected.

Best regards,